# Effects of Salicylic Acid Concentration and Post-Treatment Time on the Direct and Systemic Chemical Defense Responses in Maize (*Zea mays* L.) Following Exogenous Foliar Application

**DOI:** 10.3390/molecules27206917

**Published:** 2022-10-15

**Authors:** Yuanjiao Feng, Xiaoyi Wang, Tiantian Du, Yinghua Shu, Fengxiao Tan, Jianwu Wang

**Affiliations:** 1Key Laboratory of Agro-Environment in the Tropics, Ministry of Agriculture, South China Agricultural University, Guangzhou 510642, China; 2Guangdong Provincial Key Laboratory of Eco-Circular Agriculture, South China Agricultural University, Guangzhou 510642, China; 3Guangdong Engineering Research Center for Modern Eco-Agriculture and Circular Agriculture, Guangzhou 510642, China; 4Department of Ecology, College of Natural Resources and Environment, South China Agricultural University, Guangzhou 510642, China

**Keywords:** maize, salicylic acid, concentration effects, timing effects, defense chemicals, defense enzymes

## Abstract

Salicylic acid (SA) plays a critical role in allergic reactions of plants to pathogens and acquired systemic resistance. Thus far, although some research has been conducted on the direct effects of different concentrations of SA on the chemical defense response of treated plant parts (leaves) after at multiple post-treatments times, few research has reported on the systematic effects of non-treated parts (roots). Therefore, we examined direct and systemic effects of SA concentration and time following foliar application on chemical defense responses in maize variety 5422 with two fully expanded leaves. In the experiments, maize leaves were treated with different SA concentrations of 0.1, 0.5, 1.0, 2.5, 5.0 mM, and then, the presence of defense chemicals and enzymes in treated leaves and non-treated roots was measured at different time points of 3, 12, 24, 48, 72 h following SA foliar application. The results showed that direct and systemic effects of SA treatment to the leaf on chemical defense responses were related to SA concentration and time of measurement after spraying SA. In treated leaves, total phenolics content increased directly by 28.65% at the time point of 12 h following foliar application of 0.5 mM SA. DIMBOA (2,4-dihydroxy-7-methoxy-2H, 1, 4-benzoxazin-3 (4H)-one) content was directly enhanced by 80.56~551.05% after 3~72 h following 0.5~5.0 mM SA treatments. Polyphenol oxidase and superoxide dismutase activities were directly enhanced after 12~72 h following 0.5~5.0 mM SA treatments, whereas peroxidase and catalase activities were increased after 3~24 h following application of 1.0~5.0 mM SA. In non-treated roots, DIMBOA content and polyphenol oxidase activity were enhanced systematically after 3~48 h following 1.0~5.0 mM SA foliar treatments. Superoxide dismutase activities were enhanced after 3~24 h following 0.5~2.5 mM SA applications, but total phenolics content, peroxidase and catalase activity decreased in some particular concentrations or at the different times of measurement in the SA treatment. It can be concluded that SA foliar application at 1.0 and 2.5 mM produces strong chemical defense responses in maize, with the optimal induction time being 24 h following the foliar application.

## 1. Introduction

Plant chemical defense responses against biotic and abiotic stress can be activated by the application of exogenous hormones to the aboveground parts of plants. Examples of exogenous hormones that can enhance plant defense responses include jasmonic acid (JA), indole acetic acid (IAA), salicylic acid (SA) and others [1,2,3,4,5]. SA is one of the ubiquitous endogenous signal molecules in plants that can activate defense and protection mechanisms for disease resistance. It plays a critical role in plant allergic reactions to pathogens and systemic acquired resistance [6,7,8,9,10]. Exogenous SA application can stimulate diverse defense responses to resist adverse environmental stress and enhance the related enzyme activity in plant tissues rapidly, such as disease resistance, heavy metal resistance and water stress resistance [11,12,13,14,15]. In addition, excessive accumulation of reactive oxygen species (ROS) can cause oxidative damage in the cellular environment. The primary ROS-scavenging mechanism involves enzymatic antioxidants and metabolites. The major antioxidant enzymes include superoxide dismutase, catalase and peroxidase. Superoxide dismutase catalyzes the dismutation of O_2_^−^ to O_2_ and H_2_O_2_, which are further converted to water and oxygen by catalase or peroxidase [16]. According to some research by scholars home and abroad, defense enzymes might be involved in the synthesis of phenolics and flavonoids. Indicators of this response include the presence of phenylalnine ammonialyase, nitrate reductase, superoxide dismutase, catalase, peroxidase and polyphenol oxidase [17,18,19,20], as well as the contents of DIMBOA (2,4-dihydroxy-7-methoxy-2H, 1,4-benzoxazin-3 (4H)-one), phenolic acids, glucosinolates, caffeotannic acids and total flavonoids [2,21,22,23,24,25,26].

A mass of studies have proven that the treatment of SA to aboveground parts can affect the foliar defense response directly [2,12,21,25,27,28]. For example, foliar spraying of 3.0 mM SA on *Thymus vulgaris* L. increased the contents of total phenolics and total flavonoids in leaves after two months following the application [2]. Foliar treatment with 100 μM SA during 3~5 d in *Artemisia annua* L. significantly enhanced the activities of superoxide dismutase, catalase, ascorbate peroxidase and glutathione reductase in leaves [12]. Furthermore, 300 ppm SA mediated the aboveground growth of pomegranate (*Punica granatum* L.) and enhanced the activities of nitrate reductase, superoxide dismutase, catalase and peroxidase endogenously in leaves after 120 d [25].

Once induced in aboveground parts, the plant signal and chemical defense substances induced by the application of SA will migrate to underground parts, hence leading to the systemic plant defense responses [19,21,22,29,30]. For example, Ludwig-Müller et al. [21] found that the foliar treatment with 5.0 mM SA in Chinese cabbage (*Brassica campestris* ssp. *pekinensis*) increased the contents of total glucosinolate and 2-phenylethyl-glucosinolate in roots after 8 d post-application. In response to foliar spraying of 0.1 mM SA on *Kyoho grapevine*, the root activities of peroxidase and catalase, as well as the root content of malonyldialdehyde, were enhanced systematically after 1 d [31]. Khanna et al. [30] found that the root activities of catalase and superoxide dismutase increased by 400.0 μM SA after 2 h following the foliar treatment with SA on maize (*Zea mays* L.) LM-11 seedlings (which is relatively heat susceptible). Similarly, the foliar application of 1.0 mM SA on Oueslati olive variety (*Olea europeae* L.) resulted in an increase in total phenolics and flavonoids in root contents after 15 d [19].

Consequently, it is well known that the defense responses are induced both in leaves and roots by exogenous application of SA. In the above reports, a study has been conducted on the direct effects of different concentrations of SA on the defense response of treated plant parts (leaves) after some specific periods under certain treatments, while the systematic effects on the defense response of non-treated parts (roots) were not reported. An effective defense system against diseases and pests [32,33,34,35] has evolved in *Zea mays* L., a momentous cereal crop throughout the world. Increased concentrations of the primary defense chemicals in maize, DIMBOA and phenolic acids can prevent the invasion of pathogenic microorganisms [36,37,38]. Likewise, maize can enhance their defense effects via regulating protective enzymes [39,40]. In our study, an investigation was conducted on the contents of DIMBOA and total phenolics, the activities of polyphenol oxidase, peroxidase, catalase and superoxide dismutase in both leaves and roots of maize after 3, 12, 24, 48, 72 h following the exogenous applications of 0.1, 0.5, 1.0, 2.5, 5.0 mM SA to the aboveground parts in maize variety 5422 with two fully expanded leaves. Our hypothesis includes two aspects: (1) Treatments with different concentrations of salicylic acid on the aboveground parts after different times can directly affect the chemical defense responses of the leaves, and these reactions also have concentration and time effects. (2) Treatments with salicylic acid on maize leaves will systematically affect the chemical defense responses of roots, and have concentration and time effects. On the basis of the obtained results, the potential effects of SA concentration and time following foliar applications on direct or systemic chemical defense responses of maize were clarified.

## 2. Results

### 2.1. The Direct Effect of SA Concentration and Post-Treatment Time on Chemical Defense Responses in Treated Leaves

#### 2.1.1. DIMBOA Content

Compared with control (Figure 1), maize leaves receiving foliar spraying with SA of different concentrations showed a direct upward trend in their DIMBOA content after 3~72 h. All concentrations in SA treatment could significantly increase the content of DIMBOA in leaves after 3 and 48 h. The DIMBOA content in leaves increased significantly by 292.81%, 445.96%, 551.05% and 88.98% respectively after 24 h following the treatments of 0.5, 1.0, 2.5 and 5.0 mM SA. However, after 12 and 72 h, only the 1.0 and 2.5 mM SA treatments led to a significant increase (12 h: 80.56% and 88.98%; 72 h:134.98% and 150.57%) in the DIMBOA content in leaves.

#### 2.1.2. Total Phenolics Content

When SA of various concentrations between 0.1 and 5.0 mM was applied to aboveground maize parts, no remarkable direct impact was observed on the total phenolics content in maize leaves after 3, 24, 48 and 72 h (Figure 2). However, only 0.5 mM SA treatment caused a significant increase of 28.65% in the leaf total phenolics content after 12 h.

#### 2.1.3. Polyphenol Oxidase Activity

Following 0.1~5.0 mM SA treatments, no direct impacts were observed on the leaf polyphenol oxidase activity after 3 and 72 h. However, after 12~48 h, polyphenol oxidase activity increased under treatments of some specific SA concentrations (Figure 3). After 12 h, the leaf polyphenol oxidase activity enhanced remarkably by 146.03%, 134.86% and 117.17% under the treatments of 1.0, 2.5 and 5.0 mM SA. The leaf polyphenol oxidase activity increased by 65.55%, 159.53%, 99.19% and 64.09% after 24 h under the treatment with 0.5~5.0 mM SA. However, after 48 h, only 1.0 mM SA treatment led to a significant increase of 78.58% in the leaf polyphenol oxidase activity.

#### 2.1.4. Peroxidase Activity

Leaf peroxidase activity was dramatically enhanced after 3~24 h under the treatment with 1.0 and 2.5 mM SA, whereas the activities decreased by 42.30% after 48 h under 0.1 mM SA concentration treatments (Figure 4). When 1.0 mM SA solution was sprayed on the aboveground parts, the leaf peroxidase activity was enhanced by 257.61%, 370.93% and 451.33%, respectively, after 3, 12 and 24 h, and was also enhanced by 171.22%, 467.97% and 446.26% after 3, 12 and 24 h under the treatment with 2.5 mM SA.

#### 2.1.5. Catalase Activity

No direct effect was observed on the leaf catalase activity after 48 h under 0.1~5.0 mM SA treatments, but the catalase activity increased after 3, 12, 24 and 72 h under the treatments of some specific SA concentrations (Figure 5). After 3 h following the 2.5 mM SA treatment, the leaf catalase activity increased significantly by 172.57% compared with that of the control. The leaf catalase activity was enhanced notably after 12 and 24 h under the treatments of 1.0 and 2.5 mM SA (12 h: 389.53% and 428.83%; 72 h: 1555.56% and 1459.79%). It was also increased by 135.55% and 160.03%, respectively, after 72 h under the treatments of 2.5 and 5.0 mM SA.

#### 2.1.6. Superoxide Dismutase Activity

The foliar application of 0.5~2.5 mM SA resulted in an increase in leaf superoxide dismutase activity after 12~72 h (Figure 6). Leaf superoxide dismutase activity was enhanced by 95.26% and 66.49%, respectively, after 12 h under the treatments of 0.5 and 1.0 mM SA. After 24 h, the leaf superoxide dismutase activity increased by 71.53% and 77.92%, respectively, under the treatments of 1.0 and 2.5 mM SA. Likewise, there was an increase in leaf superoxide dismutase activity of 69.91%, 119.10% and 79.98%, respectively, after 48 h under the treatment of 0.5, 1.0 and 2.5 mM SA. However, only under 2.5 mM SA treatment, the leaf superoxide dismutase activity increased significantly by 85.98% after 72 h.

### 2.2. The Systemic Effect of SA Concentration and Post-Treatment Time on Chemical Defense Responses in Non-Treated Roots

#### 2.2.1. DIMBOA Content

The root DIMBOA content increased systematically after the induction of 0.1~5.0 mM SA to aboveground parts at 3~72 h (Figure 7). After 3 and 24 h, the 1.0 and 2.5 mM SA treatments showed a significant increase (3 h: 118.20% and 114.07%; 24 h: 85.46% and 108.03%) in root DIMBOA content. After 12 h, the root DIMBOA content increased systematically by 196.76% and 103.05% when 2.5 and 5.0 mM SA, respectively, were used to treat the leaves. Similarly, after 48 h following the treatments with 1.0, 2.5 and 5 mM SA, the root DIMBOA content increased systematically by 97.01%, 88.65% and 76.56%, respectively. In addition, following the applications of 0.1 and 2.5 mM SA, root DIMBOA content was enhanced systematically by 101.33% and 84.43% after 72 h.

#### 2.2.2. Total Phenolics Content

After 0.1~5.0 mM SA was sprayed on the aboveground parts of maize, the root total phenolics content was reduced or increased at different times (Figure 8). For example, compared with that of the control group, root total phenolics content was reduced systematically by 47.55% and 46.26% after 12 h following the applications of 0.5 and 5.0 mM SA. In addition, following the application of 5.0 mM SA, root total phenolics content was reduced by 50.81% after 24 h. However, only under the 5.0 mM SA treatment after 48 h, the root total phenolics content increased significantly by 22.46%.

#### 2.2.3. Polyphenol Oxidase Activity

Although no differences were found in the root polyphenol oxidase activity after 3 and 72 h regardless of SA concentration, a significant systemic increase was observed after 12~48 h (Figure 9). After 12 h, the treatment with 1.0 and 2.5 mM SA resulted in the systemic increase in the root polyphenol oxidase activity by 262.83% and 256.40%, respectively. Compared to that of the control group, the treatments with 1.0, 2.5 and 5 mM SA after 24 and 48 h systematically improved the root polyphenol oxidase activity (24 h: 269.29%, 242.10% and 103.89%; 48 h: 110.70%, 110.37% and 88.56%).

#### 2.2.4. Peroxidase Activity

No systemic effect was observed in the root peroxidase activity after 3, 24 and 48 h under 0.1~5.0 mM SA treatments, but peroxidase activity was increased after 12 h and was reduced after 72 h under the treatments of some specific SA concentration (Figure 10). The root peroxidase activity rose systematically by 36.06%, 41.10% and 52.33%, after 12 h under the treatments of 0.1, 1.0 and 2.5 mM SA on the aboveground parts. However, the treatment of 0.5 and 1.0 mM SA caused an obvious decrease in peroxidase activity of roots by 39.04% and 52.46% after 72 h.

#### 2.2.5. Catalase Activity

The root catalase activity can be raised systematically after 12 and 24 h following foliar applications of 1.0 and 2.5 mM SA, whereas the foliar applications of other SA concentrations (i.e., 0.5~5.0 mM) led to a decrease in root catalase activity after 72 h (Figure 11). Root catalase activity was enhanced systematically after 12 and 24 h following the application of 1.0 and 2.5 mM SA to aboveground parts (12 h: 54.19% and 90.49%; 24 h: 138.49% and 129.74%). However, the root catalase activity decreased by 37.52%, 38.56%, 40.49% and 54.25%, respectively, after 72 h following the application of 0.5~5.0 mM SA, when compared with the control group.

#### 2.2.6. Superoxide Dismutase Activity

As shown in Figure 12, a significant rise occurred in the root superoxide dismutase activity after 3~24 h under the treatments of specific SA concentrations. No systemic effects were observed in the root superoxide dismutase activity after 48 and 72 h under the 0.1~5.0 mM SA treatment. Compared to that of the control group, the treatment of 1.0 mM SA systematically improved the root superoxide dismutase activity by 128.02% after 3 h. The root superoxide dismutase activity rose systematically by 94.09%, 139.19% and 52.33%, after 12 h under the treatments of 0.5, 1.0 and 2.5 mM SA on the aboveground parts. In addition, through the application of 1.0 and 2.5 mM SA, root superoxide dismutase activity was increased systematically by 156.37% and 161.84% after 24 h.

## 3. Discussion

In the study of the direct and systematic effects of SA treatment to the aboveground parts of plants on the chemical defense responses of the treated leaves, most researchers only considered single SA treatment concentration and treatment time, and they obtained different results on the changes in the content of defense substances and the activities of defense enzymes. [2,13,25,41,42]. In our study, we confirmed that the changes in defense chemical content and enzyme activity are correlated with SA concentrations and post-treatment time, which is in line with the results of previous research on other crops [12,19,27,30,43,44,45]. As we know, photosynthates and other plant compounds are transported from aboveground parts of plants to their underground parts mainly by means of phloem vessels [46]. Therefore, the systematic effects of exogenous salicylic acid on the chemical defense responses of non-treated plant parts have also attracted the attention of some scholars [18,19,47]. For example, Methenni et al. [19] reported that the foliar application of 1.0 mM SA systematically increased the root contents of total phenolics and flavonoids in the Oueslati olive variety after 15 d. However, we reported for the first time that the concentrations of salicylic acid and post-treatment time had different effects on the same detection index of the same parts of maize seedlings. It is shown in our research results that DIMBOA content of treated leaves was directly enhanced after 3~72 h under the 0.5~5.0 mM SA treatments, polyphenol oxidase and superoxide dismutase activities were directly enhanced after 12~72 h under the 0.5~5.0 mM SA treatments, whereas peroxidase and catalase activities were increased after 3~24 h by the application of 1.0~5.0 mM SA. In non-treated roots, DIMBOA content and polyphenol oxidase activity were enhanced systematically after 3~48 h under the 1.0~5.0 mM SA foliar treatments, superoxide dismutase activities were enhanced after 3~24 h under the treatments of 0.5~2.5 mM SA, but total phenolics content, peroxidase and catalase activity decreased under treatments of some specific SA concentrations and at some specific post-treatment time.

As we know, the defense responses in gramineous plants occur via synthesis and release of defensive secondary metabolites, such as phenolic and benzoxazinoid compounds, which are the most vital direct defense chemicals against the invasion of diseases and pests [48,49]. The content of defense chemicals can be directly influenced by treating the leaves with different SA concentrations and at different post-treatment times. For instance, Ludwig-Müller et al. [21] reported that the leaf content of 2-phenylethyl-glucosinolate increased dramatically by the foliar induction of 5.0 mM SA after 8 d in wild-type cabbage. Similarly, the contents of total phenolics, chlorogenic acid and flavonoids in melon (*Cucumis melo*) leaf increased significantly after 8 d following the application of 1.0 mM SA on the plants [50]. Increased content of DIMBOA, total phenolics, coumaric acid, caffeic acid and syringic acid were found in leaves after 7 d following the 2.5 mM SA treatment on the aboveground parts of maize [22]. It was observed that the leaf MDA content increased obviously after 2.0 mg·L^−1^ SA was sprayed on the wheat (*Triticum aestivum*) [27]. The total phenolic and total flavonoid contents increased dramatically in leaves after two months following the application of 3.0 mM SA on thyme [2]. Simultaneously, there was a notable increase in leaf defense chemicals after leaves were sprayed with multiple SA concentrations. For example, a significant increase in total phenolics of *Vigna mungo* leaf was observed after 72 h following the treatment of 10.0, 50.0 and 100.0 μM SA [51]. After 20 days following the treatment of 5.0~15.0 mM SA on aboveground parts, the proline and amino acid contents were significantly increased in maize leaves [29]. What we found in our research was consistent with the above findings: total phenolic content increased directly by 28.65% after 12 h following the foliar application of 0.5 mM SA; DIMBOA content was directly enhanced by 80.56~551.05% after 3~72 h following the 0.5~5.0 mM SA treatments. However, the content of malonyldialdehyde in leaves was reduced after 9 d following the applications of 0.5 and 1.0 mM SA in *Vicia faba* L. and maize [52,53]. Kundu et al. [51] affirmed that no dramatic difference occurred in the content of total phenolics in leaves after treating *Vigna mungo* with 10.0 μM SA. These different results may reflect that there are various factors regarding experimental materials, experimental varieties, SA concentrations and experimental periods related to diverse consequences. The results of our current study confirmed that 1.0 and 2.5 mM SA treatments on maize shoots had the most obvious direct effect on the increase in the content of DIMBOA in leaves after 3~72 h.

Being a crucial component in defense systems, the synthesis of plant defense enzyme is a momentous prerequisite for defense system formation [54]. It has been verified that the defense enzymes, such as peroxidase, polyphenol oxidase, catalase and superoxide dismutase, are involved in fundamentally regulating long-term co-evolution between plants and nature [20,55]. The activity of leaf defense enzymes can be affected directly by treatments with multiple SA concentrations and different post-treatment times as found in our study. The results showed that polyphenol oxidase and superoxide dismutase activities were directly enhanced after 12~72 h following 0.5~5.0 mM SA treatments, whereas peroxidase and catalase activities were increased after 3~24 h following applications of 1.0~5.0 mM SA. These results are mostly consistent with those of previous studies [13,17,25,56,57]. For instance, according to Maity et al. [25], concentrations of nitrate reductase, superoxide dismutase, catalase and peroxidase improved the response to pomegranate plant sprayed with 300 ppm SA after 120 d. Data also revealed that the leaf defense enzymes activities were enhanced visibly when the aboveground plant parts were treated with various SA concentrations at multiple time points [5,12,27,30,51,58]. It was suggested in some study that the leaf superoxide dismutase and peroxidase activities were elevated after 36 h following the exogenous spraying of 0.5~4.0 mM SA on *Capsicum annuum* L. and after 4, 8, 12, 16, 24, 32, 36 and 48 h following the exogenous spraying of SA (0.5, 1.0, 2.0, 3.0 and 4.0 mM) on *Capsicum annuum* L. [58]. It was also shown that spraying *A. annua* with 100.0 μM SA increased the activities of superoxide dismutase, catalase, ascorbate peroxidase and glutathione reductase in leaves after 3~5 d [12]. Here, the increase in polyphenol oxidase, peroxidase, catalase and superoxide dismutase activities in maize leaves was the most evident, and it was associated to the treatments with the concentrations of 1.0 and 2.5 mM SA after 12 and 24 h. In recent years, defense priming has been reported by some researchers [59]. Priming is a newly theorized strategy of plants to protect against herbivore attacks, and it might be a vital mechanism of integrated pest control. Plant anti-herbivore and disease defense priming can be initiated by the exogenous application of organic compounds, such as SA and JA. It appears that SA can alter the evolution of disease and expel herbivores when leaves are treated directly with these compounds [25,56]. In agreement with the present study, Hegde et al. [56] showed that the increase in peroxidase and polyphenol oxidase activities in *Solanum melongena* Linn. leaves can defend against borer (*Leucinodes orbonalis* Guenee) after 4~72 h following the treatment of 1.0 mM SA. An assay was conducted to relieve the bacterial blight (*Xanthomonas axonopodis* pv. *punicae*) infection by the foliar application of SA in pomegranate [25]. Thus, it needs to be further verified whether the foliar application of SA can directly reduce the disease and control insect pests on leaves, as well as the defense effect.

Photosynthates and other plant compounds are transported from aboveground plant parts to belowground parts in virtue of phloem vessels [46]. Thus, the SA treatments on aboveground parts can not only directly affect the chemical defense response of treated parts (leaves), but can also systematically affect the chemical defense response of non-treated parts (roots) [60,61]. Our results showed that SA concentrations applied to aboveground parts and post-treatment time produced systemic root defense responses. For example, DIMBOA content was enhanced systematically after 3~48 h following 1.0~5.0 mM SA foliar treatments, which is consistent with the research results by other scholars [19,29]. The above examples [29,62] showed systemic increases in root proline, amino acid and malonyldialdehyde after 20 d following exogenous spraying of maize with 5.0~15.0 mM SA and the foliar stimulation of 1.0 g·L^−1^ SA for *Pinus wangii*, respectively. Exposure of olive seedling leaves to 0.5 and 1.0 mM SA revealed that after 15 d, the contents of total phenolics and flavonoid in roots were increased under the treatment of 1.0 mM SA [19]. However, compared with that of the control group, the total phenolics content in roots was reduced systematically by 47.55% and 46.26% after 12 h following the application of 0.5 and 5.0 mM SA. This is consistent with the research results of Feng et al. [22], who proved that the root contents of DIMBOA, coumaric acid, caffeic acid and syringic acid declined systematically after 7 d following the foliar application of 2.5 mM SA.

The treatment of aboveground maize parts under the treatments with different concentrations of SA at different post-treatment times can also systematically affect the defense enzyme activity of non-treated roots, leading to a systematic defense response. The results show that polyphenol oxidase activity of non-treated roots was enhanced systematically after 3~48 h following the foliar treatments of 1.0~5.0 mM SA, superoxide dismutase activities were enhanced after 3~24 h following 0.5~2.5 mM SA applications, but peroxidase and catalase activity decreased regarding the treatments with specific SA concentrations and specific post-treatment time. However, in contrast with the results of previous studies, we found a systemic reduction in polyphenol oxidase activity of roots after 72 h following the foliar treatment with 0.5 and 1.0 mM SA, and a systemic reduction in root catalase activity after 48 and 72 h following the post-treatments with 0.5~5.0 mM SA solution [17,31,42]. To mention the examples here, a systemic improvement was observed for root peroxidase and catalase activities after 1 d following the application of 0.1 mM SA on grape plants [31]. Ali [42] reported that the foliar application of 1 × 10^−5^ M SA in mung bean can increase the root activities of catalase, peroxidase and superoxide dismutase after 21 d. The discrepancy in the reported values for defense enzymes activities might be explained by the differences in plant genetic backgrounds, SA concentrations and post-treatment times. In addition, a systemic increase in root defense enzyme activity in response to the application of a range of SA concentrations is a common phenomenon [30,63]. For maize, 400 μM SA induced ascorbate peroxidase and glutathione reductase in CML-32 roots, and activities of catalase and superoxide dismutase in LM-11 roots showed systemic increases after 2 h following the treatments with eight SA concentrations ranging from 10 to 800 μM [30]. In consequence, our study clarified that the most apparent systemic impact on the activities of polyphenol oxidase, peroxidase, catalase and superoxide dismutase occurred in maize roots after 12 and 24 h as a result of 1.0 and 2.5 mM SA applications. Research shows that aboveground parts induced via SA may have a systemic influence on disease and pest defense in non-treated parts (root) [19,47]. Bhar et al. [47] demonstrated that the infections with *Fusarium oxysporum* of chickpea roots were decreased dramatically when the leaves were treated with 200 μM SA. Thus, further research should be conducted on whether the foliar application of SA can systematically reduce the disease and insect pests of roots, as well as the defense effect.

It is well known that aboveground and belowground parts of plants are connected closely by xylem and phloem vessels through which water, nutrients, photosynthates and other plant compounds are transported [46,64,65]. Hence, the signal compounds and defense chemicals induced by disease and pests or exogenous chemicals in belowground parts may be transmitted to aboveground parts, and accordingly, initiate the corresponding defense responses in aboveground parts. Several studies were conducted to understand the defense responses in treated roots and systemic responses in non-treated leaves by exogenous SA immersion [17,66,67,68,69,70]. For instance, Song et al. [17] showed that the activities of peroxidase, catalase and superoxide dismutase in leaves and roots were enhanced distinctly after 21 d following the treatments on roots with 0.1 mM SA immersion. Therefore, more investigations need to be completed on whether there are direct or systematic effects of exogenous salicylic acid treatments on the belowground parts and whether the chemical defense responses of the treated roots and non-treated leaves of maize seedlings have concentration and time effects. Meanwhile, other research groups have shown that the defense responses to insects and pathogens in roots are directly correlated with SA treatments [71,72], and the defense responses of antibiotic stress in leaves also has a systemic impact [41]. As a consequence, we need further studies to elucidate the influence of SA treatments on the resistance of belowground parts to disease and pests of maize roots and leaves.

## 4. Materials and Methods

### 4.1. Materials

The maize variety 5422 used in this experiment was donated by Cindy Nakatsu (Department of Agriculture, Purdue University) and was provided by Beck’s Hybrids Superior Company (Mount Pleasant, IA, USA), while SA ((±)-Salicylic acid) was purchased from the Sigma-Aldrich company (St. Louis, MO, USA). The molecular weight of solid SA is 138.12 g·mol^−1^ (99%).

### 4.2. Experimental Design

Maize seeds were surface-sterilized in 5% hydrogen peroxide solution for 5 min, before they were germinated in cheese cloth with distilled water at 25 ± 1 °C. Maize plants were grown in a controlled environment chamber (Institute of Tropical and Subtropical Ecology, South China Agricultural University) with a 12:12 h light:dark cycle. The temperature was kept at 28 and 22 °C during day and night, respectively, while the relative humidity remained at 70%. The seedlings were then transplanted to plastic pots (two seedlings per pot) with 500 mL of nutrient solution (5 mM KNO_3_, 5 mM Ca(NO_3_)_2_, 2 mM MgSO_4_, 1 mM KH_2_PO_4_, 46 μM H_3_BO_3_, 9 μM MnCl_2_, 0.8 μM ZnSO_4_, 0.3 μM CuSO_4_, 0.1 μM H_8_MoN_2_O_4_ and 20 μM FeNaEDTA) applied every two days. The treatments with SA solution were started when the maize seedlings had grown two fully expanded leaves. There were six treatments according to the concentration of SA solution, i.e., 0.1, 0.5, 1.0, 2.5 and 5.0 mM and a control (CON), with each treatment replicated four times and 0.14% ethanol and 0.05% Tween-20 contained in each solution. The treatments include evenly spraying 100 μL SA solution to the front of the two fully expanded leaves at a certain time. The same volume of distilled water with 0.14% ethanol and 0.05% Tween-20 was used as control. Leaves and roots of each plant were treated and collected at 3, 12, 24, 48 and 72 h to determine the content of DIMBOA and total phenolics, as well as the activities of polyphenol oxidase, peroxidase, catalase, and superoxide dismutase.

### 4.3. Analysis of DIMBOA

The procedure to prepare samples for DIMBOA analysis was slightly modified from Ni and Quisenberry [73]. Fresh leaves and roots were weighed and ground in 10 mL of distilled water using a mortar. Aqueous extracts were incubated for 20 min at room temperature, while samples were diluted with methyl alcohol in a ratio of 1:1. The methanol-diluted extract was centrifuged at 12,000 rpm for 15 min before filtered. The resulting filtrate was evaporated to dryness under vacuum. The residue was dissolved in 2 mL of mixed solution (acetonitrile: 0.5% aqueous acetic acid = 1:1, *v*/*v*). Then, extracts were filtered through 0.45 μm membrane filters to obtain the samples that were stored at −20 °C for further measurements.

DIMBOA concentrations were quantified by high performance liquid chromatography (HPLC) (Agilent 1100, Palo Alto, CA, USA) (column, Hypersil ODS C18 column (250 × 4 mm, 5 μm)) with diode array detector using external standard curves. Gradient elution was performed with a gradient of A (acetonitrile) and B (0.5% aqueous acetic acid), i.e., 25–45% of A from 0–10 min and 45–25% of A from 10–15 min. The solvent flow rate was set at 1 mL·min^−1^. The injection volume was 20 μL, and the detection wavelength was 262 nm. DIMBOA concentrations in leaves and roots were determined according to the standard calibration curve obtained by peak area of a series of concentrations of DIMBOA standard samples. DIMBOA standard sample was purchased from the Shanghai ACMEC biochemical technology company, whose purity is 99%.

### 4.4. Analysis of Total Phenolics

According to Randhir and Shetty [74], total phenolic contents were determined as gallic acid equivalents. The samples were weighed and ground into powder in liquid nitrogen, soaked in 10 mL of 95% ethanol, and then kept in a freezer for 48 h. After that, it was centrifuged at 12,000 rpm for 10 min and then filtered. Root: 1 mL of filtrate was transplanted in a test tube, where 1 mL of 95% ethanol, 5 mL of distilled water and 0.5 mL of Folin–Ciocalteu phenol reagent were added later. Leaf: 0.5 mL of filtrate was transplanted in a test tube, where 1.5 mL of 95% ethanol, 5 mL of distilled water and 0.5 mL of Folin–Ciocalteu phenol reagent were added later. Following the incubation for 5 min, 1 mL of 5% Na_2_CO_3_ was added, and the solutions were mixed well and kept in the dark for 1 h. The content of total phenolics was measured at 725 nm using a UV–visible spectrophotometer (UV-2450 SHIMADZU, Kyoto, Japan).

### 4.5. Determination of Polyphenol Oxidase Activity

The polyphenol oxidase crude enzyme was prepared according to Sivakumar and Sharma [75]. The samples were homogenized individually with 1 mL of 0.1 M phosphate buffer (pH 6.5) in the ratio of 1:5 (*w*/*v*), and centrifuged at 6000 rpm, at 4 °C for 15 min, whose supernatants were then used as crude enzyme for estimation.

The crude enzyme solution (10 μL), sample dilution (40 μL), and 6 concentrations (0, 15, 30, 60, 120, 180 U·L^−1^) of standard solutions were added to each well of microplate (Rapidbio Company, Plymouth, MI, USA), and thereafter, the wells were incubated for 30 min at 37 °C. After the microplate wells were washed with buffer five times, 50 μL HRP-conjugate reagent was added into the wells, which were then incubated at 37 °C for 15 min (evaded the light preservation) and washed five more times. Chromogen solution A (50 μL) and 50 μL chromogen solution B were added to each well, followed by incubation again. The reaction was stopped by adding 50 μL stop solution to each well. The absorbance at 450 nm was measured by a microplate reader. The straight line regression equation of the standard curve was calculated with the standard density and the OD value.

### 4.6. Determination of Peroxidase Activity

The activity of peroxidase in leaves and roots was quantified using the guaiacol colorimetric method described by Gao [76]. Samples (0.1 g f. wt) were homogenized with 1 mL of 0.05 mol·L^−1^ PBS (Phosphate Buffered Saline) in an ice bath and then centrifuged at 4 °C for 15 min at 8000 rpm. The supernatants were collected and used for the assay. For POD, the oxidation of guaiacol was evaluated based on the increase in absorbance at 470 nm every 30 s for 2 min. The assay contained 0.95 mL of 0.2% guaiacol, 1 mL phosphate buffer solution (pH 7.0) and 0.05 mL enzyme extract. The reaction was started with 1 mL of 0.3% H_2_O_2._

### 4.7. Determination of Catalase Activity

Catalase activity was measured in leaves and roots by hydrogen peroxide decomposition according to the method of Li [77]. Samples (0.1 g f. wt) were homogenized with 1 mL of 0.05 mol·L^−1^ PBS in an ice bath and then centrifuged at 4 °C for 15 min at 8000 rpm. The supernatants were collected and used for the assay. For CAT, the decomposition of H_2_O_2_ was followed by the decline in absorbance at 240 nm measured every 30 s for 2 min. A volume of 3 mL of reaction mixture contained 1 mL phosphate buffer solution (pH 7.0), 1 mL of 0.3% H_2_O_2_, 0.95 mL of 0.2% guaiacol and 0.05 mL enzyme extract. The reaction was initiated when the enzyme extract was added.

### 4.8. Determination of Superoxide Dismutase Activity

Superoxide dismutase activity in leaves and roots was determined by measuring its ability to inhibit the photochemical reduction of nitroblue tetrazolium according to the method of Gao [76]. Samples (0.1 g f. wt) were homogenized with 1 mL of 0.05 mol·L^−1^ PBS (pH 7.8) in an ice bath and then centrifuged at 4 °C for 15 min at 8000 rpm. Finally, 3 mL of reaction mixture was added, which contained 1.75 mL of 0.05 mol·L^−1^ PBS (pH 7.8), 0.3 mL of 130 mmol·L^−1^ methionine, 0.3 mL of 750 μmol·L^−1^ NBT, 0.3 mL of 100 μmol·L^−1^ EDTA-Na_2_, 0.05 mL enzyme extract, and 0.3 mL of 20 μmol·L^−1^ riboflavin. The test tubes containing the mixture were placed under two fluorescent lamps at 4000 lux. The reaction was started by switching on the light and allowed to run for 20 min. The reaction was stopped by switching off the light, and the absorbance at 560 nm was recorded. A non-irradiated reaction mixture was used as the control, and its absorbance was subtracted from *A*_560_ of the irradiated samples. The reaction mixture without enzyme developed maximum color as a result of complete reduction of NBT. The reduction of NBT was inversely proportional to the enzyme activity. One unit of SOD activity was defined as the amount of enzyme required to cause 50% inhibition of the rate of NBT reduction at 560 nm.

### 4.9. Statistical Analysis

The data expressed as the means ± standard errors of four replicates were obtained from each SA treatment and timing treatment as well as from controls. ANOVA analyses were performed using SPSS 25 and Origin 2018 for data arrangement. Multiple comparisons were performed with the Duncan method, with *p* < 0.05 regarded as a statistically significant level.

## 5. Conclusions

The results indicated that the SA concentration applied to leaves and post-treatment time can effectively induce the plant chemical defense reaction. In addition, direct and systemic effects of SA treatment to the leaf on the chemical defense responses were related to SA concentrations and time of measurement following the application of SA. It can be concluded that the strongest induction concentration of the SA applied to aboveground parts was 1.0 and 2.5 mM, and the optimal induction time was 24 h. Thus, a further study should be conducted on the possibility of providing direct and systemic protection for leaves and roots to achieve disease prevention and pest control after 24 h following foliar applications with 1.0 and 2.5 mM SA on maize. At the same time, the defense effect also needs to be further verified in the field.

## Figures and Tables

**Figure 1 molecules-27-06917-f001:**
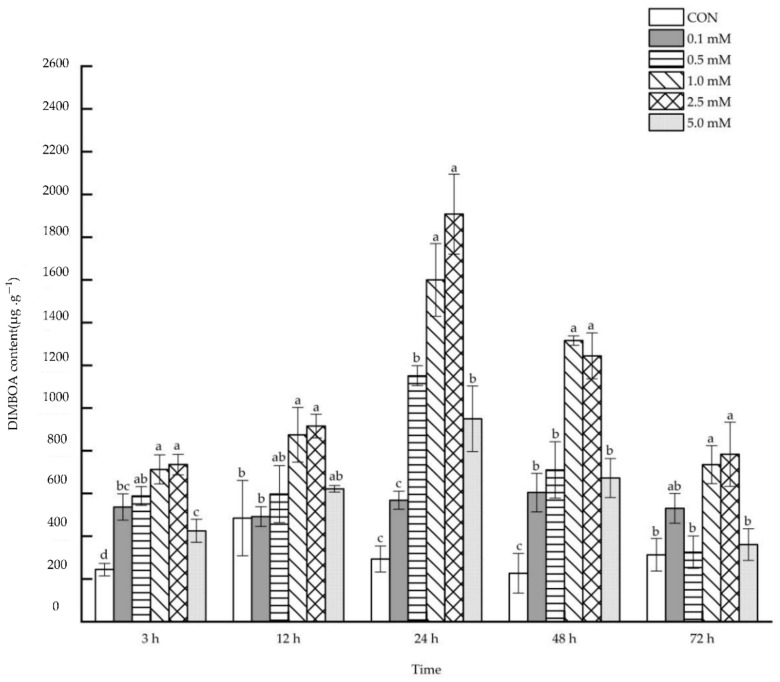
The direct effect of the SA concentration and post-treatment time on the DIMBOA content in leaves following exogenous foliar application. The data are presented as mean ± standard errors. The significance of data was determined by the Duncan method. The difference between different letters at the same time reached significant levels of 5%.

**Figure 2 molecules-27-06917-f002:**
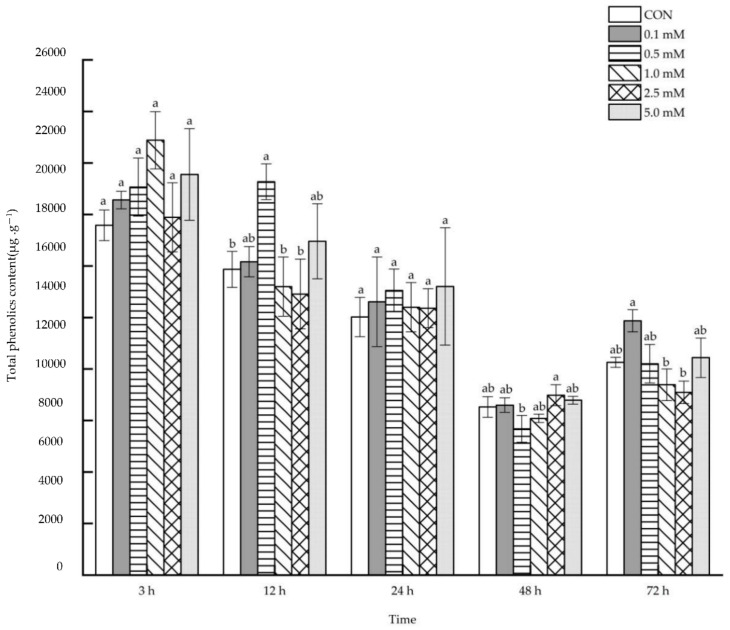
The direct effect of the SA concentration and post-treatment time on the total phenolics content in leaves following exogenous foliar application. The data are presented as mean ± standard errors. The significance of data was determined by the Duncan method. The difference between different letters at the same time reached significant levels of 5%.

**Figure 3 molecules-27-06917-f003:**
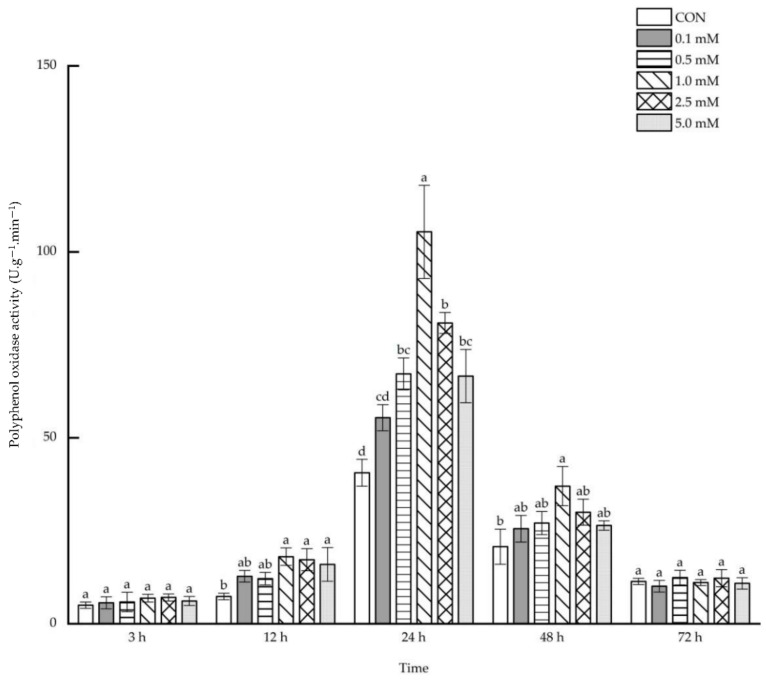
The direct effect of the SA concentration and post-treatment time on the polyphenol oxidase activity in leaves following exogenous foliar application. The data are presented as mean ± standard errors. The significance of data was determined by the Duncan method. The difference between different letters at the same time reached significant levels of 5%.

**Figure 4 molecules-27-06917-f004:**
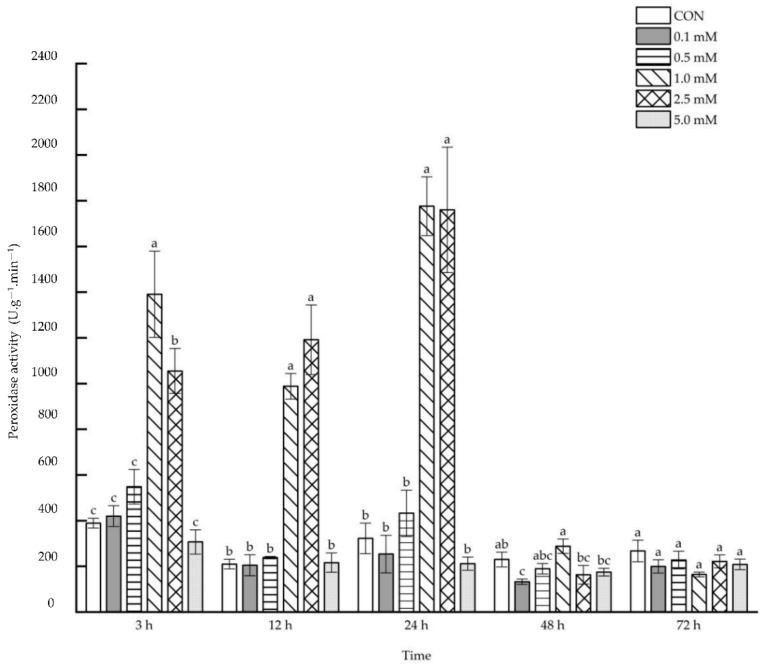
The direct effect of the SA concentration and post-treatment time on the peroxidase activity in leaves following exogenous foliar application. The data are presented as mean ± standard errors. The significance of data was determined by the Duncan method. The difference between different letters at the same time reached significant levels of 5%.

**Figure 5 molecules-27-06917-f005:**
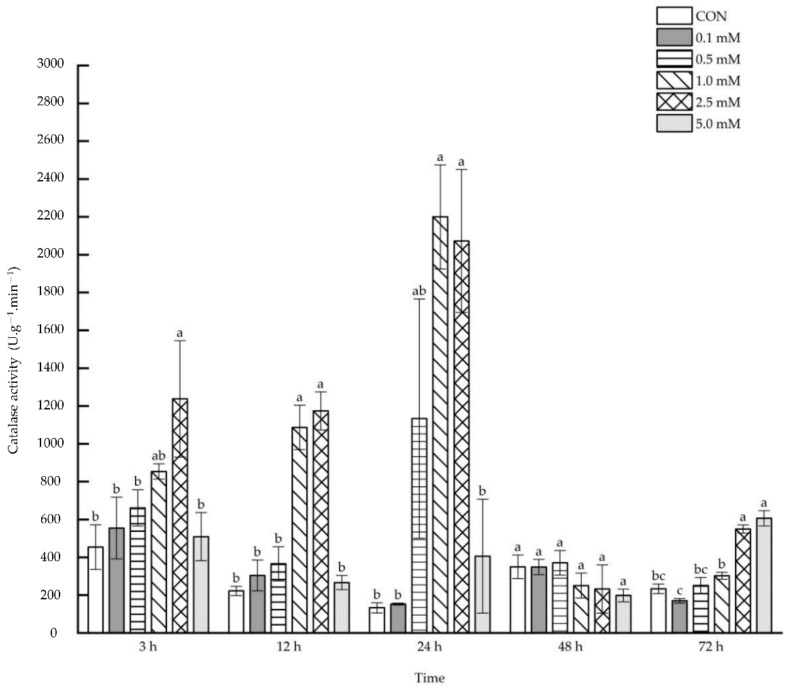
The direct effect of the SA concentration and post-treatment time on the catalase activity in leaves following exogenous foliar application. The data are presented as mean ± standard errors. The significance of data was determined by the Duncan method. The difference between different letters at the same time reached significant levels of 5%.

**Figure 6 molecules-27-06917-f006:**
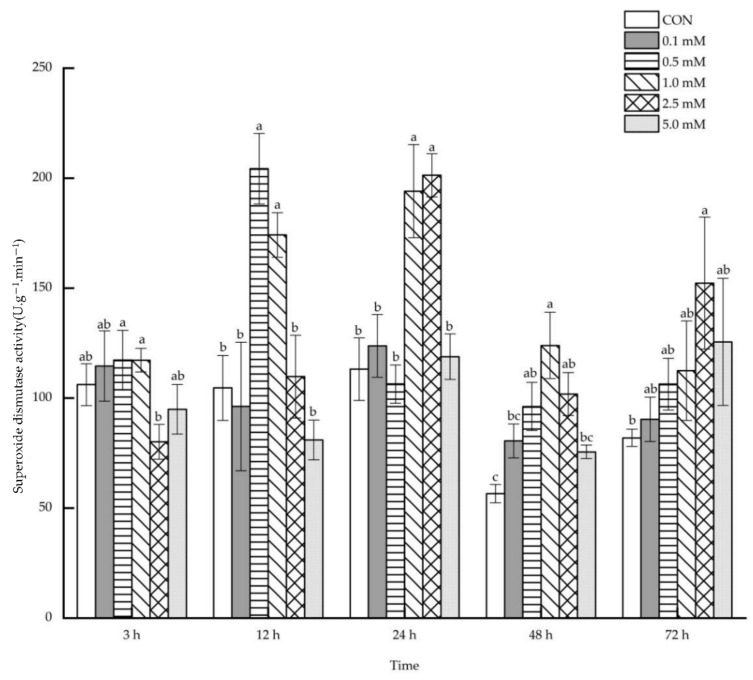
The direct effect of the SA concentration and post-treatment time on the superoxide dismutase activity in leaves following exogenous foliar application. The data are presented as mean ± standard errors. The significance of data was determined by the Duncan method. The difference between different letters at the same time reached significant levels of 5%.

**Figure 7 molecules-27-06917-f007:**
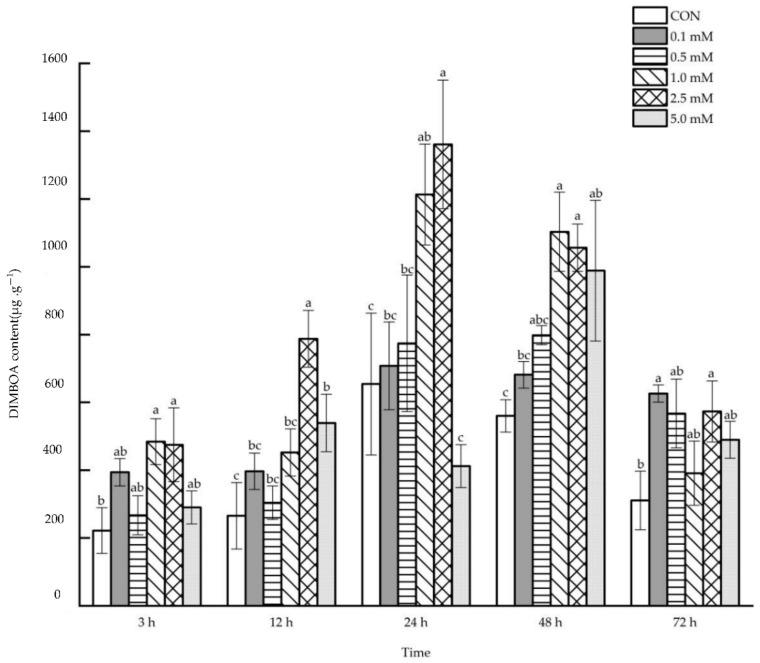
The systemic effect of the SA concentration and post-treatment time on the DIMBOA content in roots following exogenous foliar application. The data are presented as mean ± standard errors. The significance of data was determined by the Duncan method. The difference between different letters at the same time reached significant levels of 5%.

**Figure 8 molecules-27-06917-f008:**
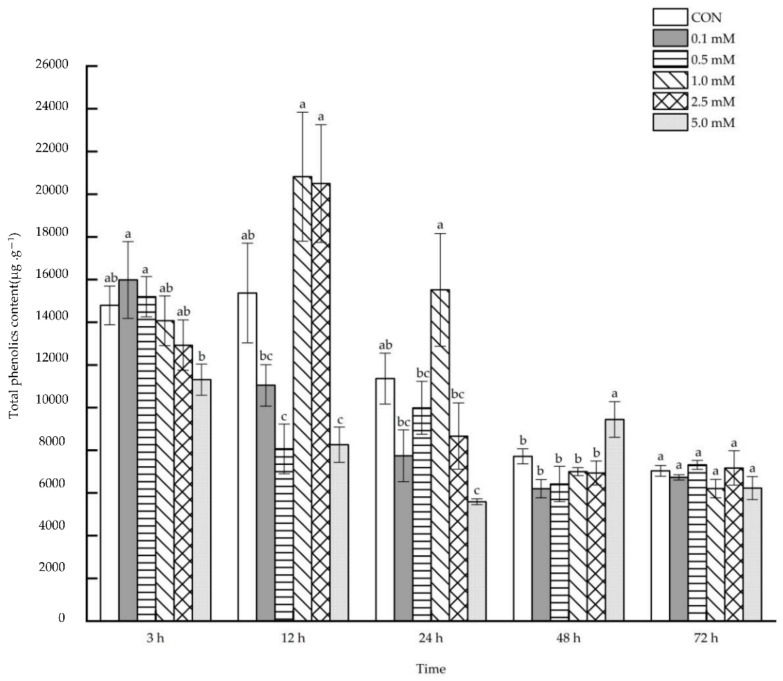
The systemic effect of the SA concentration and post-treatment time on the total phenolics content in roots following exogenous foliar application. The data are presented as mean ± standard errors. The significance of data was determined by the Duncan method. The difference between different letters at the same time reached significant levels of 5%.

**Figure 9 molecules-27-06917-f009:**
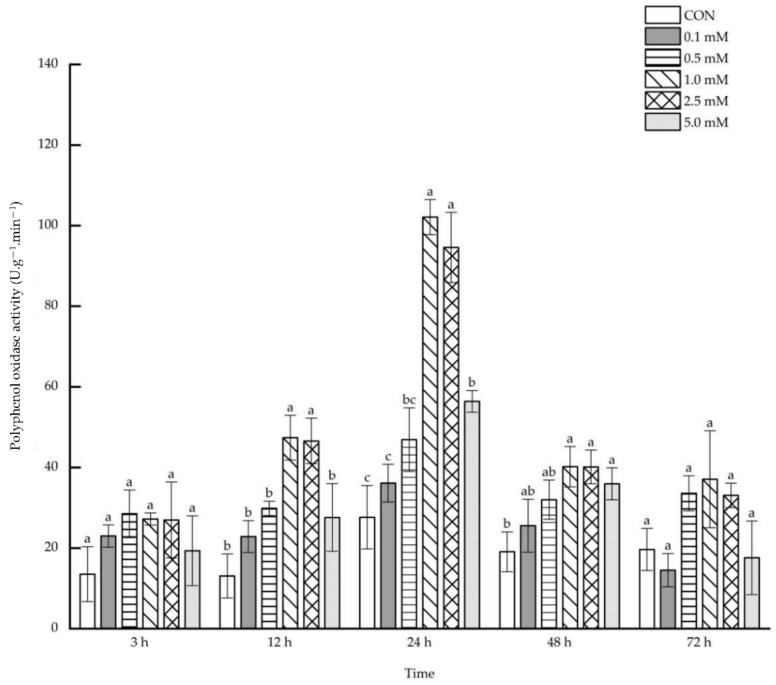
The systemic effect of the SA concentration and post-treatment time on the polyphenol oxidase activity in roots following exogenous foliar application. The data are presented as mean ± standard errors. The significance of data was determined by the Duncan method. The difference between different letters at the same time reached significant levels of 5%.

**Figure 10 molecules-27-06917-f010:**
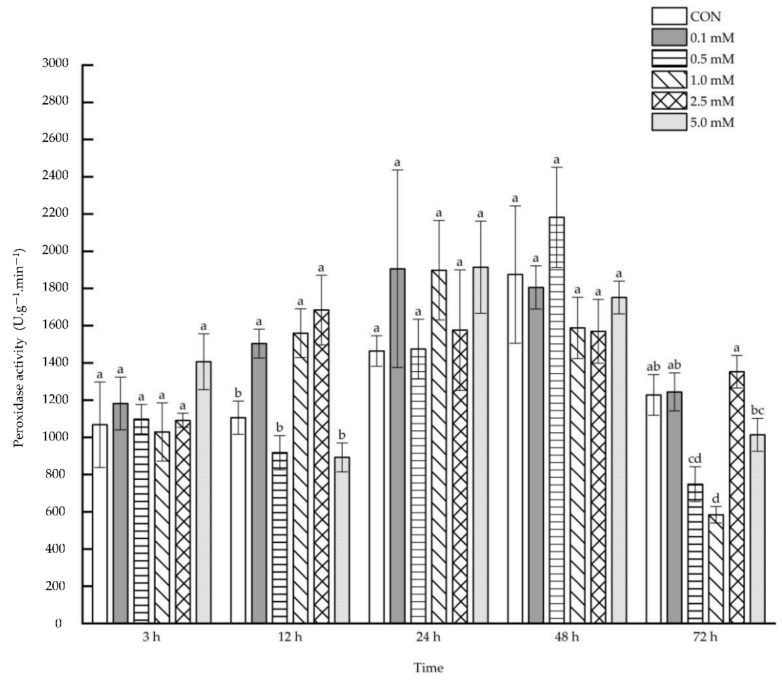
The systemic effect of the SA concentration and post-treatment time on the peroxidase activity in roots following exogenous foliar application. The data are presented as mean ± standard errors. The significance of data was determined by the Duncan method. The difference between different letters at the same time reached significant levels of 5%.

**Figure 11 molecules-27-06917-f011:**
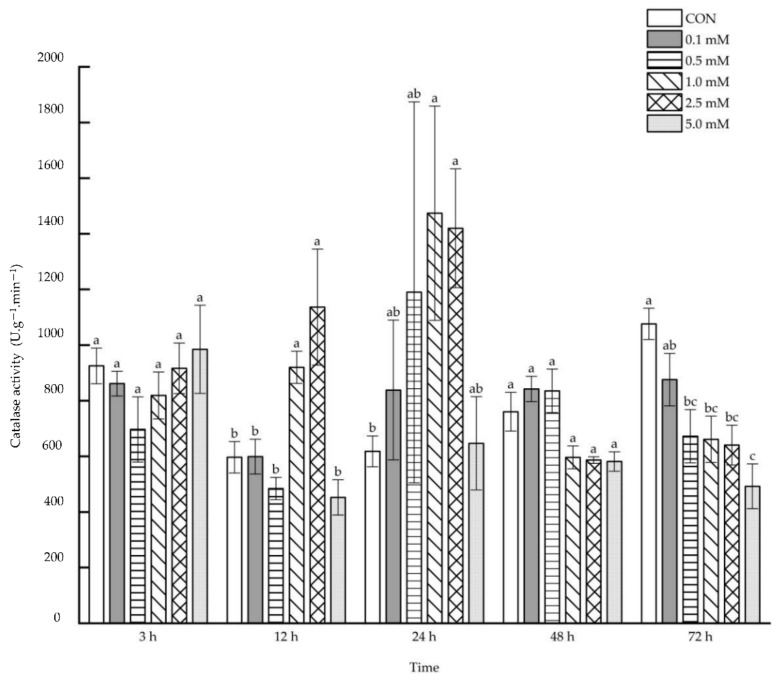
The systemic effect of the SA concentration and post-treatment time on the catalase activity in roots following exogenous foliar application. The data are presented as mean ± standard errors. The significance of data was determined by the Duncan method. The difference between different letters at the same time reached significant levels of 5%.

**Figure 12 molecules-27-06917-f012:**
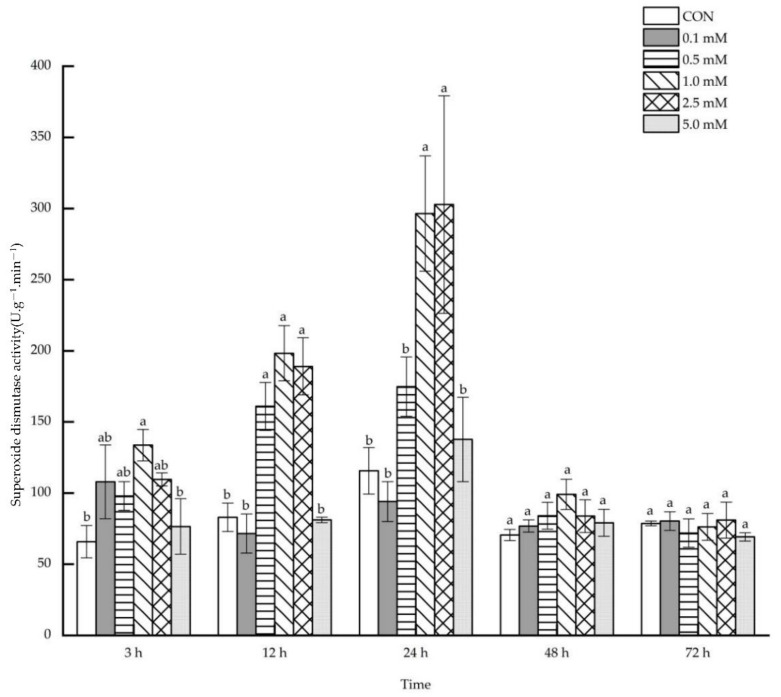
The systemic effect of the SA concentration and post-treatment time on the superoxide dismutase activity in roots following exogenous foliar application. The data are presented as mean ± standard errors. The significance of data was determined by the Duncan method. The difference between different letters at the same time reached significant levels of 5%.

## Data Availability

Data recorded in the current study are available in all Figures of the manuscript.

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
