# Peer review of "Effects of Salicylic Acid Concentration and Post-Treatment Time on the Direct and Systemic Chemical Defense Responses in Maize (Zea mays L.) Following Exogenous Foliar Application"

_molecules, 2022, doi:10.3390/molecules27206917_

Round 1
Reviewer 1 Report
Feng et al., investigated the effect of Salicylic Acid Concentration and Post-treatment Time on the defense responses of maize crops. The MS needs substantial improvement before final publication in the highly praised Molecules journal.
The abstract section is written poorly, it must start with a problem statement and then solutions. I also suggest the authors add some numerical values of their study findings in the abstract section of the manuscript.
The overuse of acronyms makes the manuscript difficult to read in many places. Therefore, I strongly recommend the authors to improve the readability of their text, starting from the abstract and all the way to the conclusions section.
Authors must add a clear hypothesis before the objectives and they must also make the objectives clear and attractive.
The quality of figures must be improved in order to meet the standards set by Molecules Journal.
The conclusion section is quite poorly written and it must be improved to give a clear message to the readers.
In the conclusion section, the authors must future research directions. What is left after this study and what should be studied in the future?
There are many mistakes in English grammar, syntax, and punctuation that distracts the readers, therefore, please carefully review your manuscript before re-submission.
Author Response
Response to Reviewer Comments
Point 1: The abstract section is written poorly, it must start with a problem statement and then solutions. I also suggest the authors add some numerical values of their study findings in the abstract section of the manuscript.
Response 1: We have revised the abstract according to the comments.
Point 2: The overuse of acronyms makes the manuscript difficult to read in many places. Therefore, I strongly recommend the authors to improve the readability of their text, starting from the abstract and all the way to the conclusions section.
Response 2: We have revised the manuscript according to the comments.
Point 3: Authors must add a clear hypothesis before the objectives and they must also make the objectives clear and attractive.
Response 3: The hypothesis has been added before the objectives
Point 4: The quality of figures must be improved in order to meet the standards set by Molecules Journal.
Response 4: We have improved all the figures in the text to meet the standards of Molecules Journal.
Point 5: The conclusion section is quite poorly written and it must be improved to give a clear message to the readers.
Response 5: The conclusion section has been improved according the comments.
Point 6: In the conclusion section, the authors must future research directions. What is left after this study and what should be studied in the future?
Response 6: We conclude the future research directions according to the suggestions.
Point 7: There are many mistakes in English grammar, syntax, and punctuation that distracts the readers, therefore, please carefully review your manuscript before re-submission.
Response 7: The manuscript has been improved according to the comments.
Reviewer 2 Report
This manuscript aims to demonstrate the use of use of salicylic acid to improve the growth performance and biochemical profile of Zea mays L. In my opinion, this work has certain value and is of interest to the readership of the journal. Hence this manuscript can be considered acceptable with a major revision by addressing following comments:
Major comments:
- They used term "significantly" in most of the sections without supporting them with statistical values or values. It is hard to interpret benefits of SA.
- In addition, there are too many inconsistent sentences and terms throughout the manuscript.
- The most critical is that even if SA could accelerate growth and biochemical profile, could it be practically applied under field conditions.
- Authors should proofread the MS thoroughly to correct typos and grammatical errors.
- Untreated control is missing. An additional control of untreated plants should be included in all experiments for comparison.
Specific comments:
Abstract should present important results (digital data) and conclusion more precisely with a limited introductory sentences
The full name of term DIMBOA should be provided
Keywords should be in chronological order
Introduction lacks relevant examples from the literature. I suggest authors to add relevant studies to support their study.
Authors should provide leaves and roots data under a single heading.
Discussion part should be improved. Current version of discussion is mere a repetition of results section. Moreover, authors mentioned their results are in line with other studies. Then, what are their contribution in the research area. I suggest authors to describe how their study is different/novel from others.
How did the authors decide the application levels of SA to Zea mays L.? Did they do a pre-experiment regarding dose optimization?
Conclusion section should be expand and improved with some promising results, future perspectives and large-scale applications.
Author Response
Response to Reviewer Comments
Point 1: They used term "significantly" in most of the sections without supporting them with statistical values or values. It is hard to interpret benefits of SA.
Response 1: Thanks for the comments. We recalculated the data and redrew the figures.
Point 2: In addition, there are too many inconsistent sentences and terms throughout the manuscript.
Response 2: We revised the text according to the comments.
Point 3: The most critical is that even if SA could accelerate growth and biochemical profile, could it be practically applied under field conditions.
Response 3: Our study reveals SA accelerates the chemical defense of maize, and this has implication for future field growth. This future research directions are proposed in the discussion section.
Point 4: Authors should proofread the MS thoroughly to correct typos and grammatical errors.
Response 4: We have carefully revised the manuscript according to the suggestions.
Point 5: Untreated control is missing. An additional control of untreated plants should be included in all experiments for comparison.
Response 5: In fact, untreated control was adopted in our study. The affecting percentage was calculated by (treatment-control) / control × 100% in the text. In the revised version, we recalculated the data and redrew the figures according to the comments.
Point 6: Abstract should present important results (digital data) and conclusion more precisely with a limited introductory sentences.
Response 6: We revised the abstract according to the comments.
Point 7: The full name of term DIMBOA should be provided.
Response 7: We have provided the full name of DIMBOA in the revision.
Point 8: Keywords should be in chronological order.
Response 8: Keywords are listed in chronological order in the revision.
Point 9: Introduction lacks relevant examples from the literature. I suggest authors to add relevant studies to support their study.
Response 9: Some relevant studies are provided in the introduction to support our study.
.
Point 10: Authors should provide leaves and roots data under a single heading.
Response 10: We redrew the figures according to the comments.
Point 11: Discussion part should be improved. Current version of discussion is mere a repetition of results section. Moreover, authors mentioned their results are in line with other studies. Then, what are their contribution in the research area. I suggest authors to describe how their study is different/novel from others.
Response 11: The discussion section has been improved according to the comments.
Point 12: How did the authors decide the application levels of SA to Zea mays L.? Did they do a pre-experiment regarding dose optimization?
Response 12: The application levels of SA to maize were set based on the former studies, and ultimately five levels were adopted in the study.
Point 13: Conclusion section should be expand and improved with some promising results, future perspectives and large-scale applications.
Response 13: The conclusion section has been improved according to the comments.
Round 2
Reviewer 1 Report
The authors have substantially improved the MS as per my suggestions. Therefore, it can be accepted for publication in its current form.
Reviewer 2 Report
The authors addressed all the comments and the paper is now ready for publication.
However, one minor correction is authors should rearrange Keywords in chronological order.